# Orthohantavirus Survey in Indigenous Lands in a Savannah-Like Biome, Brazil

**DOI:** 10.3390/v13061122

**Published:** 2021-06-11

**Authors:** Ana Cláudia Pereira Terças-Trettel, Alba Valéria Gomes de Melo, Renata Carvalho de Oliveira, Alexandro Guterres, Jorlan Fernandes, Liana Stretch Pereira, Marina Atanaka, Mariano Martinez Espinosa, Bernardo Rodrigues Teixeira, Cibele Rodrigues Bonvicino, Paulo Sérgio D’Andrea, Elba Regina Sampaio de Lemos

**Affiliations:** 1Nursing Department, Mato Grosso State University Campus Tangara da Serra, Tangara da Serra 78300-000, MT, Brazil; ana.claudia@unemat.br; 2Public Health Institute, Mato Grosso Federal University, Cuiaba 78060-900, MT, Brazil; marina.atanaka@gmail.com (M.A.); marianomphd@gmail.com (M.M.E.); 3Laboratory of Hantaviruses and Rickettsioses, Oswaldo Cruz Institute—FIOCRUZ, Rio de Janeiro 21040-900, RJ, Brazil; reoliveira@ioc.ficoruz.br (R.C.d.O.); guterres@ioc.ficoruz.br (A.G.); jorlan@ioc.fiocruz.br (J.F.); strech@ioc.fiocruz.br (L.S.P.); 4Health Secretary of State of Mato Grosso, Cuiaba 78060-900, MT, Brazil; albavaleriavia@hotmail.com; 5Laboratory of Biology and Parasitology of Wild Mammals Reservoirs, Oswaldo Cruz Institute—FIOCRUZ, Rio de Janeiro 21040-900, RJ, Brazil; brt@ioc.fiocruz.br (B.R.T.); cibelerb@inca.gov.br (C.R.B.); dandrea@ioc.fiocruz.br (P.S.D.)

**Keywords:** hantavirus infections, hantavirus cardiopulmonary syndrome, Indigenous population

## Abstract

In Brazil, the first confirmed cases of hantavirus cardiopulmonary syndrome in Indigenous populations occurred in 2001. The purpose of this study was to determine the seroprevalence of orthohantavirus infections in the Utiariti Indigenous land located in the southeastern region of the Brazilian Amazon. In December 2014 and 2015, a survey was conducted using an enzyme-linked immunosorbent assay in nine villages belonging to the Haliti–Paresí Indigenous communities. A total of 301 participants were enrolled in the study. Of the two study cohorts, the one from 2014 showed a prevalence of 12.4%, whereas the one from 2015 had a serum prevalence of 13.4%. Analysis of the paired samples of 110 Indigenous people who participated in both stages of the study enabled identification of four individuals who had seroconverted during the study period. Identifying the circulation of orthohantaviruses in the Utiariti Indigenous land highlights a serious public health problem in viral expansion and highlights the need to implement preventive measures appropriate to the sociocultural reality of these communities.

## 1. Introduction

Hantavirus cardiopulmonary syndrome (HCPS) is an acute and often lethal disease caused by orthohantaviruses harbored by rodents of the family *Cricetidae* (subfamilies *Sigmodontinae* and *Neotominae*) in the Americas [1,2,3,4,5]. Humans become infected by inhaling virus particles from infected rodent excreta and secretions [6,7,8,9]. HCPS cases occur throughout all of the American continent and reach different populations with the highest incidence in agricultural populations [10,11,12,13,14,15,16]. According to the Brazilian Ministry of Health, from 1993 to May 2019, Brazil confirmed 2134 cases of HCPS, and the state of Mato Grosso located in the southeastern region of the Brazilian Amazon reported the third highest incidence with 311 confirmed cases [16]. More than 16 genotypes of orthohantaviruses circulate throughout the Americas; six of them are associated with HCPS in Brazil and two districts in the Mato Grosso state, Castelo dos Sonhos and Laguna Negra, have identified *Oligoryzomys utiaritensis* and *Calomys callidus*, respectively, in rodent reservoirs [17,18,19].

Although HCPS was first recognized in an Indigenous Navajo community, the outbreak and the new virus were not enough to draw attention to the vulnerability of Indigenous populations and their close contact with rodent host populations [2]. Thus, despite the occurrence of HCPS cases in Indigenous populations, only a few seroprevalence studies have been carried out in Argentina, Brazil, and Paraguay [20,21,22,23,24]. In Brazil, the first confirmed cases of HCPS in Indigenous populations occurred in 2001, and since then, sporadic cases have been reported among these communities in the four states of the federation [16,25].

In 2010, a description of the first autochthonous cases of HCPS in Indigenous territories in the Mato Grosso state in the Kayabí Indigenous population area was reported [26]. Since then, the Health State Department of Mato Grosso has identified an increasing number of HCPS cases in the Indigenous people with records in different regions. The first case of the orthohantavirus infection in the Haliti–Paresí Indigenous community was described in 2014 in the Utiariti Indigenous people’s area that is located in mid-northern Mato Grosso. This area is surrounded by agricultural cities and is responsible for 75% of cases of the disease in the state [27]. In this region, important environmental changes resulting from deforestation by expansion of mechanized agricultural monoculture occurred, which causes an increase in the contact of humans with rodent hosts. The aim of this study was to determine the seroprevalence of the orthohantavirus infection in the Haliti–Paresí Indigenous community, describe some sociodemographic, epidemiological, and clinical aspects, and reinforce the importance of considering HCPS as the cause of acute febrile illness in the Indigenous communities in Brazil.

## 2. Materials and Methods

### 2.1. Study Area

The study was conducted in the villages of Haliti–Paresí located on the west side of the county Campo Novo do Parecis in both December 2014 and 2015 (Figure 1). Access to this area through the Indigenous reserve is via the MT 235 highway (constructed in 2009), which currently provides a source of income for the Indigenous people due to toll collections [27]. According to the 2016 data from the Special Indigenous Health District-Cuiabá, 327 inhabitants are distributed among nine villages belonging to Campo Novo do Parecis. The villages of Seringal/Cabeceira do Seringal, Chapada Azul, Quatro Cachoeiras, Bacaval, Morrim, Utiairiti, Sacre 2, Bacaiuval, and Wazare (Figure 2) were chosen due to the fact that they are located in the Campo Novo do Parecis area with a high incidence of HCPS cases in Brazil and a history of one case among the Haliti–Paresí but with no record of deaths among the Indigenous people from this region [28,29,30].

The Utiariti Indigenous land has native Cerrado (savannah-like) vegetation; the areas around it were deforested and, currently, are intended for a monoculture, which significantly contributes to the grain production in the Mato Grosso state, accounting for 28% of the Brazilian harvest. The Haliti–Paresí started in 2009 as an agricultural project, which enabled the planting of mechanized monocultures on 5000 acres within its territory, and since then, this technology has been used as a source of income.

### 2.2. Sample Collection

Human samples were collected in the study area in both December 2014 and 2015 from two cohorts and included visits to villages and specific approaches to the Indigenous population as described by Terças et al. [27]. Among the 327 Indigenous residents in the villages, 210 participated in the study, thus composing the 2014 cohort. In 2015, 201 Indigenous people participated; 110 were those from the year 2014 and 91 new members joined in 2015. The total number of participants in 2014 and 2015 was 301 people, including 110 paired samples.

Human serum and/or blood samples were subject to serological testing for anti-orthohantavirus IgG antibodies detection using an enzyme-linked immunosorbent assay (ELISA) based on the recombinant nucleoprotein of Araraquara orthohantavirus provided by the University of São Paulo (USP), Ribeirão Preto, according to the protocol described by Figueiredo et al. [31].

Briefly, one half of a 96-well microplate (Thermo Scientific^TM^) was coated with the recombinant N protein of Araraquara orthohantavirus (rNP ARAV), and the other half—with an *Escherichia coli* extract as the normal control antigens. Both were diluted in a 0.05 M carbonate buffer at pH 9.6 and a concentration of 2 μg/mL. The plates were kept at 4 °C overnight and then washed five times with 0.1% Tween 20 (Merck & Co., Inc., Kenilworth, NJ, USA) in phosphate-buffered saline (PBS). The microplate was then blocked with skimmed milk powder (BD Difco™) for 2 h at 37 °C. The wells were then washed and filled with 100 µL of diluted test serum samples at 1:400 dilution in PBS with 0.1% Tween 20 (Merck & Co., Inc., Kenilworth, NJ, USA). The plates were incubated for 1 h at 37 °C, washed as previously described, and 100 µL of the goat anti-human IgG peroxidase-conjugated secondary antibody (Sigma-Aldrich^®^, St. Louis, MO, USA) at 1:2000 dilution was placed in each well and incubated for 1 h at 37 °C. The plates were washed, and 100 µL of the 2,2’-azino-bis(3-ethylbenzothiazoline-6-sulfonic acid) diammonium salt (ABTS^TM^) substrate (Sigma-Aldrich^®^, USA) was added to each well and left on the well for 30 min at 37 °C. Objective readings of ELISA results were performed by determination of absorbance at 405 nm. The cut-off was determined by the mean optical density (OD) at 1:400 dilution after subtracting the OD of the negative antigen from that of the positive one. A serum dilution was considered positive if its OD was >0.3. All anti-orthohantavirus IgG-reactive human samples were further tested for IgM antibodies [31].

This study is in accordance with the national and international standards for research involving human subjects and was approved by the National Research Ethics Committee in Brazil (CONEP) under Protocol No. 819.939/2014 of 27 August 2014. Informed consent from adults and assent from children was obtained before beginning data collection during a meeting in which the objectives and methodologies of the research were presented and all ethical aspects were discussed.

## 3. Results

In this study, 301 individuals were included in the study, representing 92.04% (301/327) of the Utiariti Indigenous communities. Therefore, two cohorts were established, one in 2014 with 210 participants and another in 2015 with 201 individuals. One hundred ten Indigenous people were enrolled in both cohorts, which enabled a prospective evaluation. A serological survey showed that 35 Indigenous people had anti-orthohantavirus IgG antibodies, a prevalence of 11.6% (35/301), with 12.4% in 2014 (26/210) and 13.4% in 2015 (27/201). Of the 110 paired samples, 22 had positive results and, among them, four Indigenous people had seroconverted between 2014 and 2015. None of them reported leaving their territories during this period or showing any signs of acute febrile illness. Despite notification about cases of other infectious diseases such as dengue, zika virus, influenza, and leishmaniosis in this population, neither of these diseases was identified in these four Indigenous people. All IgG-reactive samples were subject to serological testing for identification of IgM antibodies, but no sample was reactive.

Although the distribution of men and women was similar with a slight predominance of men (51.5%), higher serum reactivity (54.3%; 19/35) was detected in women (Table 1). Ages ranged from four months to 106 years with a mean of 28.1 years (amplitude of 105 and variance of 357.24). For seroreactive individuals, ages ranged from two to 79 years with an average of 31.93 years (amplitude of 77 and variance of 387.6). The most frequently reported education level in both cohorts was elementary school; however, the presence of Indigenous people taking a degree in pedagogy, pharmacy, nursing, and nutrition is noticeable. Two hundred sixty-nine interviewees (89.6%) belonged to the Haliti–Paresí ethnicity by birth, and 32 individuals had diverse origins, including non-Indigenous (11), Rikbatsa (8), Manoki (6), Umutina (3), Nambikwara (3), and Irantxe (1).

With regard to villages, Bacaval village, the most populous village of the region with the highest number of participants enrolled in the study, presented only five seropositive residents. In Wazare village, in which 100% of the residents joined the study, 14 serum-reactive individuals (Figure 2) were found. This village is the only one with a confirmed HCPS case (2013) in the Haliti–Paresí territory.

Regarding the type of habitation, the cultural reality of the villages dominated by wooden houses (39.5%) is observed, which is common in traditional Indigenous dwellings (32.2%). Wazare, Seringal/Cabeceira do Seringal, Morrim, and Quatro Cachoeiras have only traditional homes. In Sacre 2 and Baicaiuval villages, houses are mainly brickwork buildings alongside traditional houses. In other villages, the houses are varied, that is, there are wooden, brickwork, and traditional buildings. All villages have electricity, running water, and shared bathrooms.

Figure 2 shows the other 11 serum-reactive individuals (31.4%) with their close family relationships living in Chapada Azul village. This village founded 15 years ago is the only one of the nine villages surrounded by monoculture plantations. Since 2009, all of the other villages have been surrounded with a savannah-like biome and mechanized monoculture crops that are located kilometers away from the villages. This proximity to the modified environment for large-scale production of grain may have facilitated the contact between Indigenous people and wild rodent reservoirs in addition to the emergence of a family “cluster.” Other orthohantavirus serum-reactive Indigenous people live in the villages of Seringal (1), Sacre (2), Bacaiuval (2), and Bacaval (5) and lack family connections.

Direct contact with wild rodents was reported by 45.8% of the population and 37.1% of orthohantavirus serum-reactive individuals; however, what attracts attention are the places in which the animals were observed, especially in dwellings, villages, and farm/plantations (Table 2), demonstrating the close contact the rodents have with the community. Evidence of contact with other animals for 27 Indigenous people with anti-orthohantavirus antibodies and 146 subjects of the study population was found. The most frequently mentioned animals were rhea (*N* = 36), macaws (*N* = 29), snakes (*N* = 23), deer (*N* = 13), tapirs, monkeys, boars (*N* = 10), and bats (*N* =2).

The most common signs and symptoms reported by Indigenous people in the previous 60 days were headache (*N* = 69), fever (*N* = 53), nausea (*N* = 29), diarrhea (*N* = 24), myalgia (*N* = 23), and abdominal pain (*N* = 22). Except for the Indigenous people who had severe clinical manifestations of HCPS and had to be hospitalized, the other seropositive individuals had possibly developed oligosymptomatic conditions or subclinical infections.

## 4. Discussion

According to the census conducted in 2010, 817,963 Indigenous people reside in Brazil, with 42,538 Indigenous peoples in the Mato Grosso state. This population is distributed over 65 ethnic groups and/or peoples. Among them, the Haliti-Paresí, consisting of 2022 individuals, reside in nine Indigenous lands [30,32]. In the Utiariti land located in the county of Campo Novo do Parecis, 327 individuals have reacquired their ethnicity and are now increasing their population and reassuming their Indigenous traditions [30].

This Indigenous population is surrounded by agricultural areas responsible for the largest grain production in the country in addition to the areas with the highest record of HCPS cases in the state of Mato Grosso [28]. It should be noted that 37 cases were recorded that confirmed HCPS in Indigenous people in Brazil from 2001 to 2017, and of these, 30 occurred in Indigenous people in Mato Grosso during the same period [25].

When compared to studies conducted in populations from different regions of Brazil, the identified overall seroprevalence (11.6%) in this Indigenous population is one of the highest since it ranges from 0.52% to 13.2% [33,34,35,36,37,38,39]. In fact, the seroprevalence described in this study is greater than that detected in Afro-descendent populations (quilombos) (2.82%; 9/319) in the neighboring state of Mato Grosso do Sul and also from gold mining areas in Mato Grosso (3.57%; 4/112) [40,41]. On the other hand, when compared to specific studies in Indigenous communities in South America, our results are lower than the ones from Paraguay (17%) [20,21,22,23]. In Brazil, different prevalence rates of the orthohantavirus infection have been reported, with 1.9% among the Terena Indigenous people’s community in Mato Grosso do Sul, 8% among the Enenawe Nawe in mid-northern Mato Grosso, and 51.1% among the Kayabí in northern Mato Grosso after an HCPS outbreak [26,27,28].

High rates of seroprevalence in this population may be related to the proximity to agricultural areas and the increase in interaction between the Indigenous people and farmers, including the work of the Indigenous people on monocultures crops [28,30]. Moreover, four individuals seroconverted during the study, indicating the possibility of asymptomatic or oligosymptomatic clinical forms of HCPS in this population.

The ethnic diversity found within the Utiariti territory can be explained by the unions of its members with other ethnicities and with non-Indigenous people. For instance, the Nambikwara, Rikbatsa, Irantxe, Umutina, Manoki, and Arará do Pará populations also reside in this area, in addition to non-Indigenous [30,32]; however, among the 35 seroreactive individuals, only two were non-Indigenous and two were of the Manoki ethnic origin.

Although a higher prevalence of the orthohantavirus infection in the adult male population has been well-documented in serological surveys and outbreak investigations [10,11,12,14,15,16,28,42,43], we observed a slight predominance of women (54.3%) and the presence of underage children (28.6%) among seropositive individuals. Studies conducted in apparently healthy individuals from rural and urban slum communities in Chile and Brazil have also detected orthohantivirus antibodies in individuals associated with household-related occupations (homemakers, retirees, and students) with the indoor environment as the probable site of infection [40].

In the present study, children’s ages ranged from four months to 12 years, which keeps them in areas closer to the village and can allow exposure to activities that involve playing in the house, places where food and/or garbage is stored, and handling of wild animals. It is important to highlight that the male population in the Paresí Indigenous lands represents 52.7% of the general population [30,32].

Similarly, Terças et al. [26] described the first orthohantavirus cases in the Indigenous people in the Mato Grasso territory with an outbreak in the Xingu Indigenous Park involving 18 people, six of whom progressed to the cardiopulmonary phase. A serum prevalence of 51.1% was found in the residents of this village. In this outbreak, infected children were also infected once it became evident that the risk consisted in cleaning and coming into direct contact with excreta of wild rodents at home.

Most seroreactive Indigenous people reside in the village of Wazare, which was built on the banks of the Rio Verde between 2012 and 2013. According to reports of the Indigenous population, during this period, many wild rodents invaded the houses due to deforestation to obtain wood for home construction. The only case of HCPS with a hospitalization history was one where other family members mentioned many rodents inside their residences, which may explain the large number of serum-reactive Indigenous people in this village; in addition, confirmed cases of the Indigenous people in Mato Grosso were concentrated in the regions of the Xingu Indigenous park and in the mid-northern region of the Mato Grosso state [25].

It is pertinent to consider that the expansion of agriculture to new areas and the environmental modifications required by agricultural practice, such as that recently implemented by the Haliti–Paresí, allow the entry of man into ecological niches in which contact with wild animals is intensified and new infectious agents can occur causing infections in the human population, such as the cases described here in Chapada Azul village [44,45].

Studies carried out in the state of Mato Grosso have identified reservoirs for hantavirus in the regions surrounding the Utiariti Indigenous land due to human contact with infected rodents, *O. utiaritensis* and *C. callidus*, in Castelo dos Sonhos and Laguna Negra, respectively. In addition, we suggest the infection of Indigenous people occurred as a result of contact with these rodents and associated orthohantaviruses. [18,19,46]. Although a rodent survey was conducted in the Utiariti Indigenous land in March 2015, a capture effort with 300 traps for three consecutive nights resulted in the collection of only two wild rodents, one *Cerradomys scotti* and one *Calomys tener*, with no serological evidence of orthohantavirus infections. This low number of captured rodents may be related to seasonal fluctuations in their populations. In fact, the rodent sampling period coincided with the rainy season in other Cerrado areas, a period in which some rodent species, such as *C. tener*, presents smaller population sizes [47].

Thus, it is pertinent to consider that the historical relationship between the Indigenous populations and native forests in association with the recent expansion of agriculture implemented by the Haliti–Paresí could cause constant contact with wild rodents and their infectious agents, such as orthohantaviruses [45,46]. It is also noteworthy that other reservoirs of orthohantaviruses, such as bats, were found inside homes [48,49]. Access to adequate housing is considered a basic human right and must be guaranteed to the Indigenous peoples [43] as it directly influences the health/disease process. Traditional houses in addition to wooden houses are suitable for the climate of the Cerrado (savannah-like) biome, but provide the entry of wild animals through their cracks. However, after confirming the HCPS case in 2013, the Haliti–Paresí performed architectural adaptations, including masonry materials on the floor and walls up to a height of 50 cm from the floor with the aim of restricting access of wild rodents into dwellings through the cracks between the walls and the soil.

Ultimately, the confirmation of the presence of anti-orthohantavirus antibodies in the Haliti–Paresí Indigenous communities with high prevalence rates reinforces the importance of considering HCPS as the cause of acute febrile illness in this Indigenous community since we detected evidence of viral circulation with asymptomatic or oligosymptomatic conditions or subclinical infections. The follow-up of 110 natives with a one-year interval allowed the identification of seroconversion in four natives during this period, demonstrating that orthohantaviruses are actively circulating in this community and highlighting that Indigenous lands must be constantly monitored in order to avoid occurrence or ensure early detection of HCPS cases.

## 5. Conclusions

Given the information presented in this study, the orthohantavirus infection needs to be considered in the differential diagnosis of acute febrile cases in the Indigenous populations. The need for new systematic eco-epidemiological studies in this area and monitoring of the population dynamics of wild rodents, mainly of sigmodontines, and of the orthohantavirus infection rate in this region is also evident.

Our study adds information to the limited data related to orthohantavirus infections in Indigenous communities on the American continent and point to the need for implementation of measures to improve health access via medical assistance, basic sanitation, and the sustainable use of environmental resources. These measures could help reduce the morbidity/mortality of HCPS in this population given that there still remains a disparity in local healthcare attention concerning these individuals compared to the general population.

## Figures and Tables

**Figure 1 viruses-13-01122-f001:**
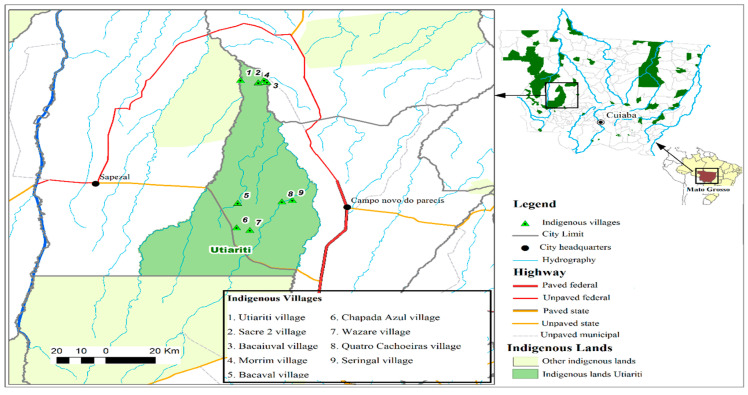
Geographical location of Mato Grosso Indigenous lands, highlighting Utiariti Indigenous land and nine villages of the Haliti-Paresí community in the southeastern region of the Brazilian Amazon, 2014–2015.

**Figure 2 viruses-13-01122-f002:**
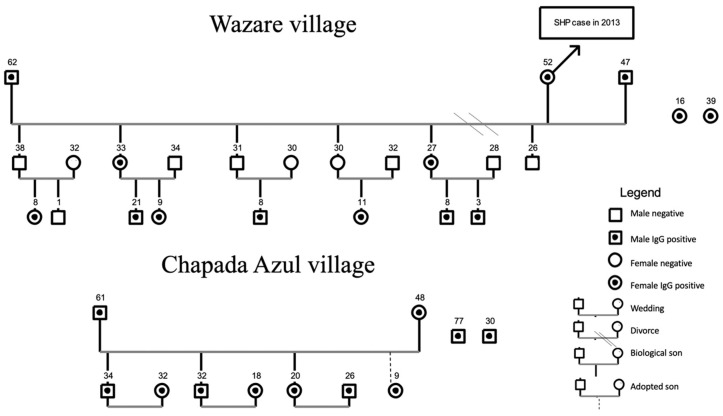
Genogram for villages of the Haliti–Paresí community, seroprevalence study of orthohantaviruses, southeastern region of the Brazilian Amazon, 2014–2015. The numbers represent the age of each person.

**Table 1 viruses-13-01122-t001:** Sociodemographic characteristics of the total population of the Haliti–Paresí Indigenous people (*N* = 301), Utiariti Indigenous land, Campo Novo do Parecis, 2014–2015.

Variables		Total Population		Anti-Hantavirus IgG+ Indigenous People	
		*N*	%	*N*	*N*
**Gender**	Male	155	51.5	16	45.7
	Female	146	48.5	19	54.3
**Education**	No education	22	7.3	1	2.9
	Not of school age	18	5.9	2	5.7
	Kindergarten	5	1.7	-	-
	Elementary school	165	54.8	20	57.1
	High school	77	25.6	10	28.6
	Higher education	14	4.7	2	5.7
**Ethnicity**	Paresí	269	89.4	31	88.6
	Non-Indigenous people	11	3.7	2	5.7
	Other Indigenous ethnicity	21	6.9	2	5.7
**Village of residence**	Bacaval	63	20.9	5	14.3
	Wazare	56	18.6	13	37.2
	Bacaiuval	48	15.9	3	8.6
	Utiariti	39	13	-	-
	Seringal/Cabeceira do Seringal	29	9.6	1	2.9
	Chapada Azul	27	9	11	31.4
	Quatro Cachoeiras	22	7.3	-	-
	Sacre 2	16	2.3	2	5.7
	Morrim	1	0.3	-	-
**Type of habitation**	Wood	119	39.5	11	31.4
	Traditional Indigenous housing	97	39.5	14	40
	Brickwork	85	28.2	10	28.6

**Table 2 viruses-13-01122-t002:** Clinical and epidemiological history of the total Indigenous Haliti-Paresí population (*N* = 301), Utiariti Indigenous land, Campo Novo do Parecis, 2014–2015.

Variables		Total Population		Anti-Hantavirus IgG+ Indigenous People	
		*N*	%	*N*	%
**Contact with wild rodents**	No	163	54.2	22	62.9
	Yes	138	45.8	13	37.1
	- Inside the house	42	30.4	7	53.9
	- In the village	24	17.4	5	38.5
	- On the farm	15	10.9	1	7.7
	- On the soy/corn plantation	7	5.1	2	15.4
	- Other sites	37	26.8	1	7.7
**Contact with other wild animals**	No	155	51.5	8	22.9
	Yes	146	48.5	27	77.1
**Signs and symptoms reported 60 days before serum collection**	Headache	69	22.9	5	14.3
	Fever	53	17.6	5	14.3
	Nausea	29	9.6	1	2.9
	Diarrhea	24	8	3	6.6
	Myalgia	23	7.6	7	20
	Abdominal pain	22	7.3	2	5.7
	Dizziness	15	5	2	5.7
	Low back pain	14	4.7	4	11.4
	Dyspnea	13	4.3	2	5.7
	Asthenia	11	3.7	-	-
	Cough	10	3.3	-	-
	Hypotension	6	2	-	-
	Chest pain	5	1.7	1	2.9
**Other diseases in the past 60 days**	Influenza	3	0.9	-	-
	Dengue	1	0.3	-	-
	Cutaneous Leishmaniosis	1	0.3	-	-
	Zika virus	1	0.3	-	-

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
