# Peer review of "Orthohantavirus Survey in Indigenous Lands in a Savannah-Like Biome, Brazil"

_viruses, 2021, doi:10.3390/v13061122_

Round 1

Reviewer 1 Report

The manuscript entitled “ORTHOHANTAVIRUS SURVEY IN INDIGENOUS LANDS IN THE SAVANNAH-LIKE BIOME,  BRAZIL” is complete.

The authors conducted the survey of  Orthohantavirus infections among the Haliti-Paresí indigenous community during 2014 and 2015 by using ELISA. There finding revealed that the population has been exposed to orthohantavirus. Therefore, it is important to consider the regular monitor of hantavirus exposers among the populations to avoid outbreaks in the future.

Author Response

Mr. Reviewer,

We are grateful for the valuable contributions in the revision of the manuscript and attached a document with the inclusion of all suggestions, as well as a corrected article attached to the journal's system for its evaluation.

The authors

Reviewer 2 Report

The results of this investigation provide additional knowledge on a hantavirus seroprevalence in the indigenous human populations in Brazil. However, new information presented in the manuscript is limited. In addition, this manuscript needs significant revisions to improve its style and English language, as it is certainly too much work for the Journal Editorial team. Please, see my specific comments below.

  1. English language used by the authors, both routine and scientific, is certainly suboptimal. On the first look, it seems to be OK; however, more one reads - more questions appear. It includes use of particular words, expressions and entire sentences which are not routinely used by the native English speakers. Examples are too numerous to consider listing them here. In particular,  in English language literature it is hard to find the word "indigenous" alone, it is usually used in the stable expression "indigenous people". Some sentences are difficult to understand because of the way they are written. This reviewer suspects that some automatic translation service was used by the authors, with some (insufficient!)manual editing afterwards. In fact the entire manuscript has to be carefully rewritten.
  2. Abstract has to be improved. In particular, the words "Background", "Methods", "Results" and "Conclusions" have to be removed. The first sentence of Abstract has to be deleted, and the last sentence - to be rewritten for clarity.
  3. This particular investigation has nothing to do with genetics, so Figure 2 does not add any useful information to the manuscript and can be safely deleted.
  4. Discussion section is repetitive and lacks clarity. It certainly should be rewritten.

Author Response

(The authors gave the same response as above.)

Reviewer 3 Report

I see that this is an interesting report that increases our knowledge about hantavirus cardiopulmonary syndrome (HCPS) in the Americas. I do not want to repeat the main findings of the study. Interesting findings were a high seropositivity also in children as well as obvious asymptomatic or oligosymptomatic cases.

I am a clinician. I am not an expert to comment the suggestions presented about the relationship between the main findings of the study and the changes in forests and agriculture in the study area.

Author Response

(The authors gave the same response as above.)

Reviewer 4 Report

The manuscript by Pereira eta al conducted a survey using enzyme-linked immunosorbent assay in nine

villages belonging to Haliti-Paresí indigenous community. From a total of 301 participants around 12% were seropositive and 4 individuals seroconverted during the study. Identification of circulating hantavirus Utiariti indigenous lands highlights a serious public health problem and suggest a need to implement the preventive measures appropriate to these communities to prevent the deadly disease in future. 

Suggestions:

  1. The case incidence in this area should be compared to other parts of the country to get the idea about the rate of spread.
  2. Line 70: Correct 2014
  3. Line 90: Proved the reason why some people didn’t participate in the study. Sine HCPS is deadly and there seems to be seroprevalence, it would be interesting to know if any hantavirus deaths were reported in this area
  4. The ELISA protocol used (ref. 31) should be briefly mentioned. Shorten the background information.
  5. The header on table 1 is confusing, how much is the total population to get the percentage values reported.
  6. Fig 2 legends: pleas clarify the legends, the squares and circles are connected, what does that mean
  7. Table 2: Provide the numerical number for the total population at the top of the table

Author Response

(The authors gave the same response as above.)

Round 2

Reviewer 4 Report

The reviewers have answered all my concerns. I feel the manuscript should be accepted for publications.